

# Skeletal pathology and variable anatomy in elephant feet assessed using computed tomography

Sophie Regnault[1], Jonathon J.I. Dixon[1], Chris Warren-Smith[1,2], John R. Hutchinson[1] and Renate Weller[1]

[1] Royal Veterinary College, Hertfordshire, United Kingdom
[2] Langford Veterinary Services, University of Bristol, Bristol, United Kingdom

## ABSTRACT

Foot problems are a major cause of morbidity and mortality in elephants, but are underreported due to difficulties in diagnosis, particularly of conditions affecting the bones and internal structures. Here we evaluate post-mortem computer tomographic (CT) scans of 52 feet from 21 elephants (seven African *Loxodonta africana* and 14 Asian *Elephas maximus*), describing both pathology and variant anatomy (including the appearance of phalangeal and sesamoid bones) that could be mistaken for disease. We found all the elephants in our study to have pathology of some type in at least one foot. The most common pathological changes observed were bone remodelling, enthesopathy, osseous cyst-like lesions, and osteoarthritis, with soft tissue mineralisation, osteitis, infectious osteoarthriti, subluxation, fracture and enostoses observed less frequently. Most feet had multiple categories of pathological change (81% with two or more diagnoses, versus 10% with a single diagnosis, and 9% without significant pathology). Much of the pathological change was focused over the middle/lateral digits, which bear most weight and experience high peak pressures during walking. We found remodelling and osteoarthritis to be correlated with increasing age, more enthesopathy in Asian elephants, and more cyst-like lesions in females. We also observed multipartite, missing and misshapen phalanges as common and apparently incidental findings. The proximal (paired) sesamoids can appear fused or absent, and the predigits (radial/tibial sesamoids) can be variably ossified, though are significantly more ossified in Asian elephants. Our study reinforces the need for regular examination and radiography of elephant feet to monitor for pathology and as a tool for improving welfare.

Corresponding author
Sophie Regnault, sregnault@rvc.ac.uk

# INTRODUCTION

Elephants not only provide education and entertainment as zoological attractions, but also have ecological significance as umbrella (or keystone) species, whose conservation indirectly protects others (*Choudhury et al., 2008*). They also have economic importance as tourist attractions and working animals. Welfare of elephants is an active area of discussion, both in professional fields and in general society. Although the welfare of captive elephants has been improving through husbandry initiatives and advances in knowledge of veterinary care for these species, there remain several areas that continue to be obstacles to optimum welfare.

Pathological foot conditions are one such problem area, thought to constitute the single most important health problem of captive elephants, with up to 50% of elephants in captivity suffering from foot problems, although the actual prevalence of carious conditions remains unknown (*Fowler, 2006*). Accurate diagnosis is challenging, treatment is expensive and time-consuming (*Lewis et al., 2010*) and chronic unresponsive conditions of the feet are a major reason for euthanasia in captivity (*Csuti, Sargent & Bechert, 2008*).

Some foot problems are visible externally (e.g., solar pad or cuticle lesions), do not require diagnostic imaging, and seem to be improving with the near-universal adoption of daily examination and foot care routines in elephants (*Lewis et al., 2010*). However, other pathological lesions—particularly those affecting the osseous structures—are challenging to identify and monitor. Originally superficial lesions may lead to further problems through ascending infection, resulting in osteomyelitis and/or infectious arthritis. Osteoarthritis (OA, also called degenerative joint disease/DJD) is commonly encountered and other problems are described.

Management conditions are thought to be the one of the most important factors in the development of distal limb osseous pathologies (*Fowler, 2006*; *Miller, Hogan & Meehan, 2016*). Osteomyelitis and septic arthritis are generally an extension of a soft tissue infection or penetrating solar trauma. Hard floors, lack of exercise, and repeated concussive forces (potentially including stereotypic behaviour; *Haspeslagh et al., 2013*) have all been proposed to contribute to the development of OA (*Hittmair & Vielgrader, 2000*) or general musculoskeletal foot health (*Miller, Hogan & Meehan, 2016*) . Additionally, the conformation of the large and relatively straight limbs of elephants may predispose them to pathology (*Fowler, 2006*), as might the inherent biomechanics of the feet. Pathological changes have been speculated to occur more frequently in regions that normally experience high pressures (i.e., mechanical stresses) during walking; namely the distal structures of the lateral digits (*Panagiotopoulou et al., 2012*).

Lameness is not always an obvious feature in elephants with foot problems (*Lewis et al., 2010*), and radiography of the distal limb has been described to diagnose and monitor foot problems (e.g., *Hittmair & Vielgrader, 2000*; *Siegal-Willott et al., 2008*; *Kaulfers et al., 2010*; *Mumby et al., 2013*). Over the recent years advanced imaging modalities such as computed tomography (CT) and magnetic resonance imaging (MRI) have been more commonly used in veterinary practice for musculoskeletal and other problems, but their use for elephants is precluded by body size and transport issues. As a result of the limited availability of imaging, the frequencies of these bony conditions in captive elephants are unknown and they are almost certainly under-reported based on what we know in other large animals such as cows (*Nigam & Singh, 1980*; *Kofler, Geissbühler & Steiner, 2014*) or rhinoceroses (*Regnault et al., 2013*; *Galateanu et al., 2013*).

The aims of this study were to identify pathological bone lesions in the feet of captive African (*Loxodonta africana* Blumenbach 1797) and Asian (*Elephas maximus* Linnaeus 1758) elephants using post-mortem CT. We hypothesise that when there is pathological change, it will be present in multiple feet of the same individual and also that there will be multiple kinds of pathological change, which may be due to shared predisposing factors (e.g., management conditions, as above) and/or altered use. By exploring the locations of

pathological changes, we further hypothesise that foot regions typically exposed to high pressures (i.e., lateral digits) are predisposed to developing lesions. When assessing any structures for pathology it is essential that the clinician is aware of normal anatomical variation, therefore, we also describe other osseous features that likely represent non-pathological, variable distal limb anatomy.

## MATERIALS AND METHODS

CT scans of 52 cadaver feet (16 right fore, 12 left fore, 14 right hind, 10 left hind) from 21 captive elephants (seven African *Loxodonta africana*, and 14 Asian *Elephas maximus*) were evaluated for evidence of pathology. All elephants were adult or near-adult: ranging from 17 to 61 years old. Feet or CT scans were donated to the Royal Veterinary College from various sources (zoos and safari parks) in the European Union. Data on morbidity and mortality was later compiled from an online database (http://www.elephant.se/) as well as from donating institutions, and details on the individual elephants are summarised in Table 1.

The following distal limb structures were assessed on the CT scans for all five digits (denoted DI to DV by convention); the carpometacarpal (CMC) or tarsometatarsal (TMT) joints, metapodial (metacarpal/metatarsal) bones, paired proximal sesamoids, metacarpophalangeal (MCP) or metatarsophalangeal (MTP) joints, proximal and distal interphalangeal (PIP and DIP) joints, phalangeal bones, and surrounding soft tissues. Lesions were identified and interpreted by a large animal veterinary radiologist and resident (J.D. and R.W.), and categorised in consensus using an established scheme previously used for elephants and rhinoceroses (*Regnault et al., 2013*). This grading scheme is provided in Table 2. Severity of each lesion was graded as slight, moderate, or severe (grades 1, 2 or 3 respectively; see Table 2 for grading criteria).

The degree of ossification of "predigits" (prepollex/prehallux, or radial/tibial sesamoids; e.g., *Hutchinson et al., 2008*; *Hutchinson et al., 2011*) was also noted, and categorised as: non-ossified (code 0), minimally ossified (code 1), moderate ossification embedded in (presumably) cartilaginous soft tissue (code 2), or extensively ossified single structure (code 3). Anatomical variability in the proximal sesamoid bones was described.

For analysis, each pathology category was expressed as the number of affected structures per foot e.g., if osseous cyst-like lesions were observed only in metacarpals III and IV, the foot would have two affected structures. For the more frequently observed pathological categories (remodelling, enthesopathy, osseous cyst-like lesions and osteoarthritis), a generalised estimating equation (GEE) was used to test age, sex, foot type (fore or hind), and species (Asian or African) as predictors on the amount of observed pathology (modelled as count data with a negative binomial distribution). The models ran as multi-variable negative binomial regressions with backwards selection. For statistical assessment, significance was set at $p = 0.05$. Multiple feet from the same elephant were treated as repeated measures. Similar GEE models were run for sesamoid fusion, and atypically-shaped and multipartite phalanges (though only with Asian elephants for the latter, as no African elephants had multipartite phalanges). A GEE (ordinal logistic) model was also used to test whether

Table 1 **Details of seven African (*Loxodonta africana*) and 14 Asian (*Elephas maximus*) elephants in this study.** Asterisks indicate elephants known to have foot or locomotor problems. 'Feet scanned' indicates how many feet had available CT scan data, 'Reason for death/euthanasia' details the cause of death (from donating institutions or the online database http://www.elephant.se/).

| Elephant | Feet scanned | Reason for death/euthanasia | Sex | Age (years) |
|---|---|---|---|---|
| African1 | 4 | ? | M | 19 |
| African2 | 4 | Euthanasia (vaginal/urogenital tract disease) | F | 24 |
| African3 | 1 | ? | M | 27 |
| African4 | 1 | Disease (infection, gastrointestinal, unspecified mechanical abnormality) | M | 28 |
| African5 | 1 | ? | F | 30 |
| African6 | 4 | Disease (suspected cardiac disease) | F | 32 |
| African7 | 2 | Disease (unspecified) | M | 32 |
| Asian1 | 2 | ? | M | 17 |
| Asian2* | 1 | Euthanasia (forelimb lameness) | M | 17 |
| Asian3* | 4 | Euthanasia (arthritis and aggression) | F | 26 |
| Asian4 | 3 | ? | F | 40 |
| Asian5* | 4 | Euthanasia (foot abscess) | F | 35 |
| Asian6 | 2 | ? | M | 40 |
| Asian7* | 1 | Euthanasia (chronic arthritis) | F | 40 |
| Asian8 | 3 | ? | F | 42 |
| Asian9* | 2 | Disease (osteomyelitis and foot disease) | F | 52 |
| Asian10 | 2 | Euthanasia (unspecified illness) | M | 50 |
| Asian11 | 1 | Euthanasia (unspecified) | F | 50 |
| Asian12 | 4 | Euthanasia (unspecified) | F | 55 |
| Asian13 | 2 | Sudden collapse | F | 61 |
| Asian 14 | 4 | ? | ? | ? |

**Notes.**
  M, male; F, female; ?, unknown.

species was a significant predictor of degree of predigit ossification (modelled as categorical data), and then separately within each species as bi-variable models to test if age and foot type were significant predictors. Statistical analyses were performed in IBM SPSS Statistics for Windows (Version 24.0).

To examine whether elephants with pathological lesions in one foot were more likely to have lesions in other feet, we compared the proportion of elephants with one vs. two or more feet diagnosed with pathology (only for the 15 elephants with scans of multiple feet, and pathology in at least one foot) for all categories.

# RESULTS

## Pathological changes

All of the elephant feet in this study (i.e., all adults and near-adults) were observed to have pathology of some type under our grading scheme. However, the majority of these lesions (63%) were grade 1, thus considered to be clinically insignificant or anatomical variants. We considered lesions of grade 2 or 3 (moderate and marked/severe) likely to

**Table 2 Grading scheme used for categorising pathological changes in this study.**

| Lesion type | Changes observed | Severity |
|---|---|---|
| Mineralisation | Mineral opacity within soft tissues at a site distant to other osseous structures | Slight = solitary short linear foci, occasionally coalescing Moderate = multiple linear or irregularly shaped mineral attenuating areas Severe = extensive mineralisation, frequently linear coalescing mineral structures, elongated |
| Osteitis | Disruption of normal trabecular bone pattern, mottled appearance, multiple hypoattenuating foci, loss of parts of bone, destruction of normal bone outline, periosteal new bone formation | Slight/Moderate/Severe based on extent of changes |
| Enthesopathy | Discrete new bone formation at attachment sites of tendons and ligaments | Slight/Moderate/Severe: based on size and extent of the mineral attenuation at the site of the soft tissue structures insertion onto the bone, if multiple sites affected in the same bone then interpretation based on all affected sites for overall grade. |
| Cyst-like lesions | Well-defined radiolucencies (with hyperattenuating rim) | Grade based on size (not measured), small/medium/large (observer experience-based only) |
| Fractures | Sclerotic linear areas, may be with new bone formation at bone surface (old), linear hypoattenuation (acute) | Not graded (just present/absent) |
| Osteoarthritis | Discrete new bone at periarticular surface, subchondral bone sclerosis, narrowing or obliteration of joint space, subchondral lysis, widening of joint space | Mild: small pointed periarticular osteophytes, mild increased bone attenuation or thickening of the subchondral bone plate Moderate: Multiple medium sized periarticular osteophytes, evidence of widening or narrowing of the joint space not considered to be related to limb position only, thickening of the subchondral bone and adjacent increased mineral attenuation. Severe: Numerous and extensive periarticular osteophytes, marked narrowing of the articular space, marked subchondral bone thickening/hyperattenuation. |
| Infectious arthritis | Florid new bone formation at periarticular surface, subchondral bone lysis, widening of joint space, subchondral bone sclerosis, narrowing or obliteration of joint space | Slight/Moderate/Severe based on extent of changes |
| Remodelling | Enlargement of vascular channels and synovial fossae, irregular contour to the osseous structures away from the joint surfaces and not considered entheseophyte formation, sometimes deep excavations in the bone, alterations in shape of a bone. | Subjective scale of the overall shape of the bone, degree of periosteal change identified, alterations in the cortices. No fixed categorical variables. |
| Subluxation | Loss of articular surface contact between the bones forming a joint | Not graded (just present/absent) |

represent clinically significant pathology. Based on this assessment, only grade 2 and 3 lesions were analysed further below. Forty seven of 52 feet (21/21 elephants) were found to contain pathological changes graded moderate (2) or greater. Percentages are reported for descriptive purposes.

The most frequent change observed was remodelling, especially observed as bone surface irregularities (Figs. 1A and 1D), representing 31% of all pathologies observed (see Table 3 for breakdown). Remodelling was present in 18 out of 21 elephants (39/52 feet). Commonly remodelled bones were the metapodials (with 31% of all remodelling observed

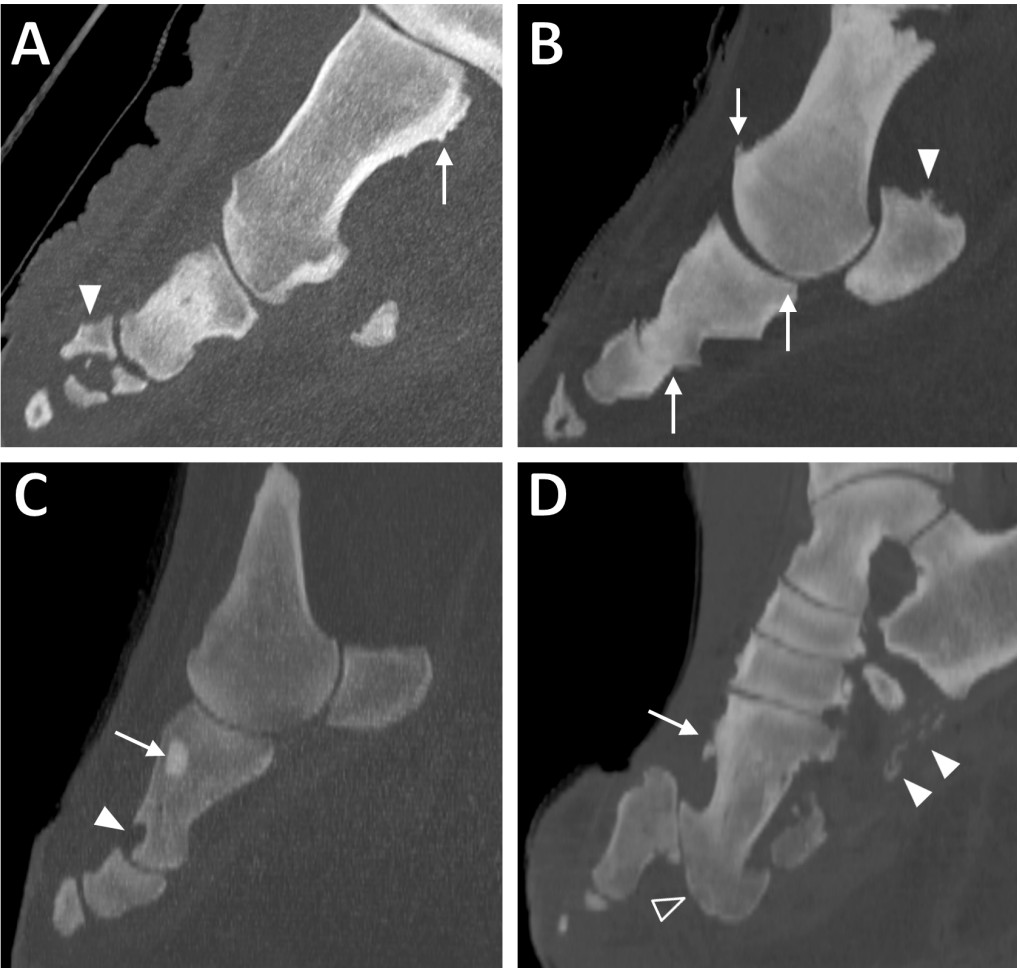

**Figure 1  Sagittal CT slices of digits in elephant feet, exhibiting pathological changes.** (A) Remodelling of the metacarpal (arrow) and fracture of the middle phalanx (filled arrowhead) in DIV of the right hind foot of 'Asian8'. (B) Enthesopathy of the proximal sesamoid (filled arrowhead) and evidence of DJD (osteophytes, altered joint spacing) at the proximal and middle interphalangeal joints (arrows) in DIV of the right forefoot of 'Asian10'. (C) Focal hyperattenuating region (arrow) and misshapen, scalloped proximal phalanx (filled arrowhead) in DII of the right forefoot of 'Asian13'. (D) Remodelling of the bones (arrow), subluxation of the proximal interphalangeal joint (unfilled arrowhead) and soft tissue mineralisation (filled arrowheads) in DIII of the right hind foot of 'Asian4'.

here), proximal phalanges (30%), sesamoid bones (16%) and middle phalanges (8%). Commonly affected digits were DIII (27% of remodelling), DIV (25%), DV (21%) and DII (17%), whilst DI appeared least affected (10%). A GEE (negative binomial model) found that observed remodelling increased with age ($p = 0.01$ in the final univariate model); age remained significant ($p = 0.03$) after accounting for species ($p = 0.2$), sex ($p = 0.8$), and foot type (fore vs. hind; $p = 0.7$) in the multivariable modelling. For the affected elephants with multiple feet scanned, remodelling was commonly observed in multiple feet (10/13 elephants with two or more affected feet, with only three elephants having a single foot affected).

Regnault et al. (2017), *PeerJ*, DOI 10.7717/peerj.2877

**Table 3 Summary of Grade 2+ pathological lesions detected in this study.** In the first column, "Af" and "As" with numbers correspond to our elephant subjects from Table 1; also "Path," number of unique pathology categories observed per individual elephant, and asterisks indicate elephants known to have foot or locomotor problems. Second column: "Foot": LH, left hind; LF, left fore; RH, right hind; RF, right fore.

| Elephant | Foot | Calcification | Osteitis | Enthesophyte | Cyst | Fracture | OA | Infectious OA | Remodelling | Subluxation | Misc. |
|---|---|---|---|---|---|---|---|---|---|---|---|
| Af1 Path: 2 | LF | 0 | 0 | 1 | 1 | 0 | 0 | 0 | 0 | 0 | 0 |
| | LH | 0 | 0 | 1 | 0 | 0 | 0 | 0 | 0 | 0 | 0 |
| | RF | 0 | 0 | 0 | 0 | 0 | 0 | 0 | 0 | 0 | 0 |
| | RH | 0 | 0 | 0 | 0 | 0 | 0 | 0 | 0 | 0 | 0 |
| Af2 Path: 2 | RH | 0 | 0 | 0 | 1 | 0 | 0 | 0 | 0 | 0 | 0 |
| | RF | 0 | 0 | 1 | 0 | 0 | 0 | 0 | 0 | 0 | 0 |
| | LF | 0 | 0 | 1 | 1 | 0 | 0 | 0 | 0 | 0 | 0 |
| | LH | 0 | 0 | 0 | 0 | 0 | 0 | 0 | 0 | 0 | 0 |
| Af3 Path: 7 | RH | 6 | 3 | 7 | 3 | 0 | 6 | 2 | 8 | 0 | 0 |
| Af4 Path: 1 | RF | 0 | 0 | 0 | 0 | 0 | 0 | 0 | 0 | 0 | 1 |
| Af5 Path: 5 | LF | 4 | 0 | 6 | 6 | 0 | 7 | 0 | 9 | 0 | 0 |
| Af6 Path: 5 | LF | 0 | 0 | 1 | 3 | 0 | 2 | 0 | 3 | 0 | 0 |
| | LH | 0 | 0 | 0 | 3 | 0 | 0 | 0 | 1 | 0 | 0 |
| | RF | 0 | 0 | 1 | 5 | 0 | 0 | 0 | 0 | 0 | 0 |
| | RH | 0 | 1 | 2 | 6 | 0 | 1 | 0 | 3 | 0 | 0 |
| Af7 Path: 3 | RF | 0 | 0 | 0 | 0 | 0 | 0 | 0 | 0 | 0 | 0 |
| | LF | 0 | 0 | 2 | 3 | 0 | 0 | 0 | 3 | 0 | 0 |
| As1 Path:3 | RF | 0 | 0 | 0 | 1 | 0 | 0 | 0 | 1 | 1 | 0 |
| | LF | 0 | 0 | 0 | 0 | 0 | 0 | 0 | 0 | 0 | 0 |
| As2* Path: 6 | RH | 2 | 0 | 8 | 1 | 1 | 1 | 0 | 8 | 0 | 0 |
| As3* Path: 4 | LF | 0 | 0 | 8 | 2 | 0 | 0 | 0 | 9 | 0 | 0 |
| | LH | 0 | 0 | 0 | 0 | 0 | 0 | 0 | 2 | 0 | 0 |
| | RF | 0 | 0 | 8 | 1 | 0 | 1 | 0 | 4 | 0 | 0 |
| | RH | 0 | 0 | 3 | 3 | 0 | 4 | 0 | 2 | 0 | 0 |
| As4 Path: 8 | LF | 0 | 1 | 4 | 3 | 0 | 3 | 0 | 4 | 0 | 0 |
| | RF | 9 | 0 | 7 | 3 | 0 | 4 | 0 | 4 | 0 | 0 |
| | RH | 6 | 5 | 10 | 11 | 0 | 9 | 2 | 12 | 1 | 0 |
| As5* Path: 4 | LF | 0 | 0 | 2 | 0 | 0 | 1 | 0 | 1 | 0 | 0 |
| | LH | 0 | 0 | 9 | 1 | 0 | 2 | 0 | 6 | 0 | 0 |
| | RF | 0 | 0 | 9 | 1 | 0 | 0 | 0 | 6 | 0 | 0 |
| | RH | 0 | 0 | 4 | 0 | 0 | 0 | 0 | 3 | 0 | 0 |

Regnault et al. (2017), *PeerJ*, DOI 10.7717/peerj.2877

**Table 3** (*continued*)

| Elephant | Foot | Calcification | Osteitis | Enthesophyte | Cyst | Fracture | OA | Infectious OA | Remodelling | Subluxation | Misc. |
|---|---|---|---|---|---|---|---|---|---|---|---|
| As6 Path: 3 | LF | 0 | 0 | 2 | 1 | 0 | 0 | 0 | 0 | 0 | 0 |
| | RF | 0 | 0 | 2 | 0 | 0 | 0 | 0 | 1 | 0 | 0 |
| As7* Path: 6 | RF | 0 | 2 | 4 | 1 | 0 | 1 | 1 | 5 | 0 | 0 |
| As8 Path: 7 | LH | 0 | 0 | 12 | 6 | 0 | 7 | 0 | 12 | 0 | 0 |
| | RF | 3 | 2 | 7 | 2 | 0 | 3 | 1 | 5 | 0 | 0 |
| | RH | 0 | 0 | 4 | 4 | 0 | 3 | 0 | 7 | 0 | 0 |
| As9* Path: 8 | LH | 6 | 1 | 3 | 2 | 0 | 2 | 1 | 6 | 1 | 0 |
| | RH | 0 | 0 | 3 | 0 | 0 | 3 | 0 | 4 | 1 | 0 |
| As10 Path: 4 | RF | 0 | 0 | 12 | 3 | 0 | 10 | 0 | 20 | 0 | 0 |
| | RH | 0 | 0 | 2 | 1 | 0 | 0 | 0 | 3 | 0 | 0 |
| As11 Path: 2 | RH | 0 | 0 | 0 | 3 | 0 | 0 | 0 | 4 | 0 | 0 |
| As12 Path: 6 | LF | 1 | 0 | 6 | 0 | 0 | 3 | 0 | 9 | 2 | 0 |
| | LH | 4 | 0 | 5 | 0 | 0 | 1 | 0 | 8 | 1 | 0 |
| | RF | 2 | 0 | 13 | 2 | 0 | 3 | 0 | 13 | 0 | 0 |
| | RH | 2 | 0 | 2 | 1 | 0 | 0 | 0 | 5 | 1 | 0 |
| As13 Path: 9 | LH | 3 | 0 | 5 | 3 | 1 | 1 | 0 | 4 | 0 | 0 |
| | RF | 1 | 0 | 7 | 6 | 0 | 6 | 1 | 8 | 1 | 2 |
| As14 Path: 7 | LF | 3 | 4 | 9 | 5 | 0 | 6 | 3 | 14 | 0 | 0 |
| | LH | 0 | 0 | 3 | 3 | 0 | 1 | 0 | 5 | 0 | 0 |
| | RF | 3 | 2 | 7 | 4 | 0 | 6 | 2 | 11 | 0 | 0 |
| | RH | 0 | 0 | 1 | 7 | 0 | 1 | 0 | 4 | 0 | 0 |
| **Total:** | | **55** | **21** | **204** | **113** | **2** | **98** | **13** | **237** | **9** | **3** |
| **755 observations** | | **(7%)** | **(3%)** | **(27%)** | **(15%)** | **(0.3%)** | **(13%)** | **(2%)** | **(31%)** | **(1%)** | **(0.4%)** |

The second most commonly identified pathology was enthesopathy (Fig. 1B), representing 27% of all pathologies observed (Table 3). Enthesopathy was present in 18/21 elephants (43/52 feet). Commonly affected regions were the metapodial bones (32%), proximal phalanges (27%), sesamoids (21%) and CMC/TMT joints (18%). Commonly affected digits were DIII (27%), DIV (24%), DV (23%) and DII (19%), whilst DI appeared least frequently affected (6%). A GEE (negative binomial model) found enthesopathy was more commonly observed in Asian compared to African elephants ($p = 0.001$ in the final univariate model); species remained significant ($p = 0.03$) after accounting for age ($p = 0.3$), sex ($p = 0.6$), and foot type ($p = 0.8$) in the multivariable modelling. For the affected elephants with multiple feet scanned, enthesopathy was almost always observed in multiple feet (13/14 elephants with two or more affected feet versus one elephant with only a single foot affected).

Osseous cyst-like lesions of bone (Figs. 2A and 2B) represented 15% of all pathologies observed (Table 3), present in 20/21 elephants (39/52 feet). Commonly affected structures were the metapodial (56%) and proximal phalangeal bones (28%). Commonly affected digits were DIV (27%), DIII (24%), DII (21%) and DV (19%), whilst DI appeared least affected (10%). A GEE (negative binomial model) found that osseous cyst-like lesions were more commonly observed in females compared to males ($p = 0.01$ in the final univariate model); sex remained significant ($p = 0.03$) after accounting for species ($p = 0.3$), age ($p = 0.5$) and foot type ($p = 0.2$) in the multivariate modelling. For the affected elephants with multiple feet scanned, osseous cyst-like lesions were generally observed in multiple feet (10/15 elephants with two or more affected feet, versus five elephants with only a single foot affected).

Osteoarthritis (OA; Fig. 1B) represented 13% of all pathologies observed (Table 3), present in 14/21 elephants (28/52 feet). Commonly affected joints were the carpometacarpal/tarsometatarsal joints (46%), metacarpophalangeal/metatarsophalangeal joint (36%), and proximal interphalangeal joint (10%). Commonly affected digits were DIII (28%), DIV (25%), DII (24%) and DI (12%), whilst DV appeared least affected by OA (11%). A GEE (negative binomial model) found that OA increased with age ($p = 0.02$ in the final univariate model); age remained significant ($p = 0.05$) after accounting for foot type ($p = 0.6$), sex ($p = 0.6$), and species ($p = 0.9$) in the multivariate modelling. For the affected elephants with multiple feet scanned, OA was almost always observed in multiple feet (9/10 elephants with two or more affected feet, versus one elephant with only a single foot affected).

Soft tissue mineralisation (Figs. 1D and 2C) represented 7% of all pathologies observed (Table 3), present in 9/21 elephants (17/52 feet). These mineralisations were identified having similar interdigital, frequently linear structure in all limbs. For the affected elephants with multiple feet scanned, mineralisation was generally observed in multiple feet (4/6 elephants with two or more feet affected, versus two elephants with only a single foot affected).

Osteitis (Fig. 2D) represented 3% of all pathologies observed (Table 3), present in 7/21 elephants (9/52 feet). Commonly affected regions were the proximal and middle phalanges (33% and 29% of observations, respectively), metapodials (24%), and sesamoids (14%). Commonly affected digits were DIV (48% of osteitis observed here), DIII (38%),

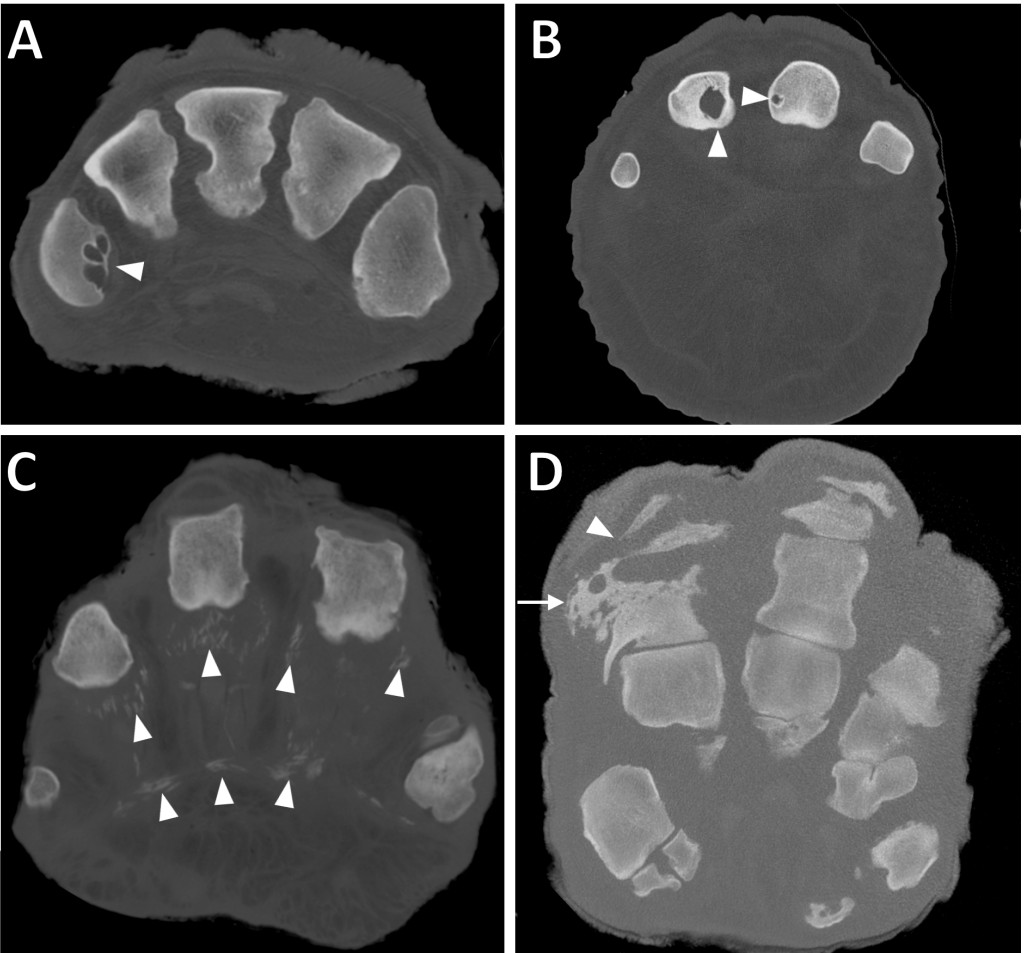

**Figure 2  Transverse CT slices of digits in elephant feet, exhibiting pathological changes.** (A) Multiple osseous cyst-like lesions in metacarpal (filled arrowhead) in DV of the right hind foot of 'African2.' (B) Solitary osseous cyst-like lesions in the proximal phalanges (filled arrowheads) of DIII and DIV of the left forefoot of 'African6.' (C) Soft tissue mineralisation on the palmar aspect of digits (filled arrowheads) in the right forefoot of 'Asian4'. (D) Osteitis of the proximal phalanx (arrow) and infectious osteoarthritis of the proximal interphalangeal joint (filled arrowhead) in DIV of the left forefoot of 'Asian14.'

whilst DV (10%) and DII (5%) appeared least affected. DI was not affected in any limb studied. For the affected elephants with multiple feet scanned, osteitis was observed roughly equally affecting multiple feet versus just one foot (2/5 elephants versus three elephants, respectively).

Infectious osteoarthritis (Fig. 2D) represented 2% of all pathology observed (Table 3), present in 7/21 elephants (8/52 feet), or 13 joints in total. In 7/8 feet, bone(s) adjacent to the affected joints were also observed with osteitis. Commonly affected joints were the MCP/MTP (46%), PIP (38%) and DIP joints (15%). Commonly affected digits were DIV (54%), DIII (38%) and DV (8%). DI and DII were unaffected in any limb. For the affected elephants with multiple feet scanned, infectious OA was generally only observed in one foot (5/6 elephants with a single affected foot, versus only one elephant with multiple feet affected).

Subluxation (Fig. 1D) of a joint represented 1% of all pathology observed (Table 3), present in five out of 21 elephants (8/52 feet). The MCP/MTP, PIP and DIP joints were equally affected. Digits were also fairly equally affected. For the affected elephants with multiple feet scanned, subluxation was observed roughly equally affecting multiple feet versus just one foot (two elephants versus three elephants, respectively). Complete luxation was not observed in any joint in this study.

Fractures (Fig. 1A) represented <1% of all pathology observed (Table 3), present in only 3/21 elephants (3/52 feet). Two of the fractures were identified in the distal phalanx of DIII, and one was of the middle phalanx of DIV.

In addition to the categories of pathology listed in Table 2, we observed focal hyperattenuating (i.e., highly dense) regions within the medullary cavities of long bones (Fig. 1C) in two out of 21 elephants (2/52 feet). Three hyperattenuating regions were observed in total: two in the metacarpals of digit III (different feet of different elephants), and one in the proximal phalanx of digit II.

In this study, multiple types of pathology were identified in most feet: out of 52 feet, two were observed with all nine pathological categories listed in Table 2, two feet with eight categories, three feet with seven categories, seven feet with six categories, 12 feet with five categories, six feet with four categories, three feet with three categories, and eight feet with two categories. Only three feet were observed with a single category of pathology, and six feet (11.5% of limbs) had no evidence of pathology.

## Anatomical variations

In the CT images evaluated, the configuration of the proximal sesamoid bones was variable: they were sometimes present as a pair, commonly fused together (appearing as a single bone), and occasionally absent from scans altogether (i.e., not visible as either an ossified bone or as an obvious soft tissue structure; Figs. 3A and 3D).

In digit I, the sesamoids often had the appearance of a single bone (42/52 feet); very occasionally they appeared as a fused pair (3/52 feet), and in only one foot appeared as an unfused pair. The digit I sesamoids were always present in African elephants, but were sometimes missing in the hind feet of Asian elephants (absent in 6/14 Asian elephants, or 8/35 hind feet).

In our sample of African elephants, the sesamoid bones in the other digits were almost always paired; only two feet out of 17 had fused sesamoids (in digits III and IV in one hind foot, and digit V in another elephant's forefoot). In Asian elephants the appearance of sesamoids in the other digits varied much more. In digit II, 22 were fused, 12 were paired, and one appeared single. In digit III, 26 were fused, eight were paired, and one was lytic and difficult to assess. In digit IV, 24 were fused, 10 paired, and one absent. In digit V, 12 were fused, 22 paired, and one appeared single. In both species, the lateral sesamoid of digit V was sometimes appreciably larger than the medial sesamoid (Fig. 3C). A GEE (negative binomial model) found that species was a statistically significant predictor ($p < 0.0005$ in both the multivariate and final univariate model) of amount of sesamoid fusion (i.e., number of fused pairs per foot, not distinguishing which pairs), with Asian elephants

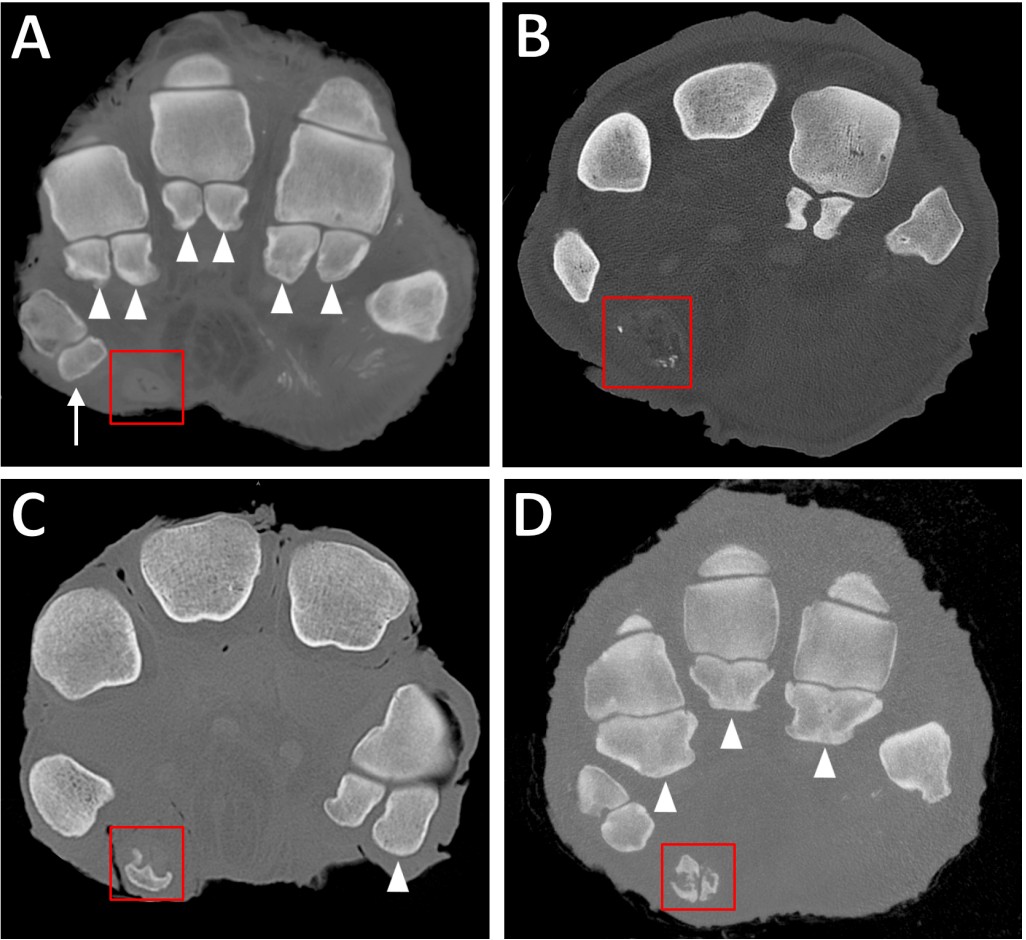

**Figure 3** **Transverse CT slices of elephants' feet, showing the sesamoids.** (A) Completely unossified prepollex (red box) in the right forefoot of 'Asian4.' Note also the single sesamoid of DI (arrow) and the paired proximal sesamoids of other digits (filled arrowheads). (B) Sparsely mineralised prepollex (red box) in right forefoot of 'African6.' (C) Medium-sized, discrete ossification of the prepollex (red box) in right forefoot of 'African2.' Note also the larger lateral sesamoid of DV (filled arrowhead) compared to the medial sesamoid. (D) Large ossification bounding the outer edges of the prepollex (red box) in right forefoot of 'Asian12.' Often, the middle of the predigit will remain partially unossified resulting in a rod-like appearance. Note also fusion of the paired proximal sesamoids (filled arrowheads) in DII–DIV, compared to the unfused sesamoids in (A).

possessing more fused sesamoids than African elephants. Sex ($p = 0.9$), foot type ($p = 0.4$), and age ($p = 0.7$) were not significant.

Ossified predigits (i.e., radial/tibial sesamoids associated with digit I) were more frequently identified in Asian than African elephants. In African elephants, 9/17 feet (3/7 elephants) had evidence of ossified predigits, compared to 27/35 feet (13/14 elephants) in Asian elephants. The extent of ossification was lower in African elephants: seven predigits were minimally ossified and two had intermediate ossification, versus one minimally ossified predigit, six with intermediate ossification, and 20 extensively ossified predigits in Asian elephants. Figure 3 shows the different degrees of predigit ossification observed. A GEE (repeated measures ordinal logistic model) found that species was a statistically

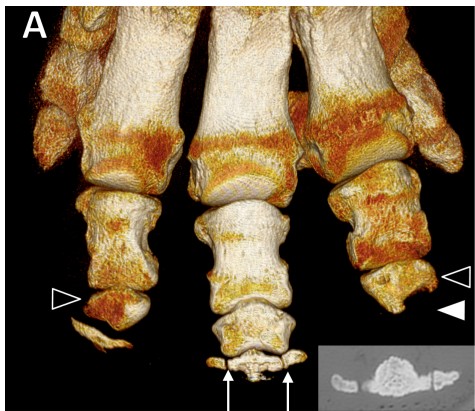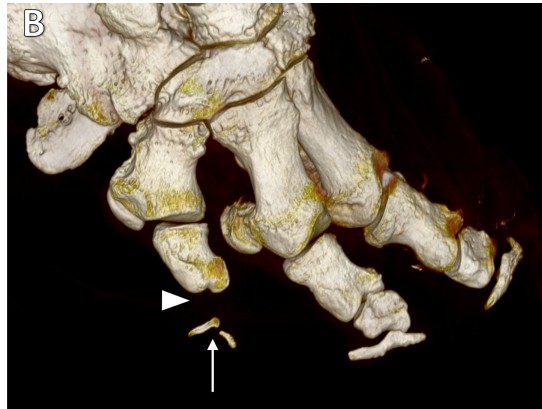

**Figure 4** **Three-dimensional reconstructions from CT scans.** (A) Dorsal view of the left forefoot of 'Asian5', showing tripartite distal phalanx of DIII (arrows; also CT appearance inset) and misshapen middle phalanges of DII and DIV (unfilled arrowheads). The middle phalanx of DII is wedge shaped, whilst that of DIV is wedged-shaped with a scalloped distal aspect and missing distal phalanx (filled arrowhead). (B) Dorso-lateral view of the right hind foot of 'Asian9' showing the bipartite distal phalanx (arrow) and missing middle phalanx (filled arrowhead) of DV.

significant predictor of presence and extent of predigit ossification ($p = 0.009$). Within each species, neither age ($p < 0.9$ in African elephants and $p = 0.5$ in Asian elephants) nor foot type (fore versus hind; $p < 0.9$ for African elephants and $p = 0.7$ for Asian elephants) were found to be statistically significant predictors of predigit ossification.

We observed multipartite distal phalanges (Fig. 4) in 36 digits of 23 feet (12 elephants; all Asian). Most were bipartite (27/36), but some were tripartite (9/36). Multipartite distal phalanges were most frequently identified in DV (16/36), DIII (9/36), DIV (6/36), and DII (5/36). DI had none. A GEE (negative binomial model) found that, within Asian elephants, neither age, sex nor foot type were statistically significant predictors of multipartite distal phalanges ($p = 0.3$, $p = 0.1$, $p = 0.1$ respectively).

We observed 25 atypically shaped phalanges in 17 feet of 11 elephants (10 Asian and one African). Affected bones were most often middle phalanges (23/25 bones), but one proximal and one distal phalanx were also observed to have atypical shapes. The shape of the bones varied, but most appeared wedge-shaped (Fig. 4A) due to relative shortening of the bone's abaxial aspect and/or mediolateral narrowing (18/25 bones). Others appeared very rounded with loss of the typical rhomboidal shape (5/25 bones), and occasionally bones had a scalloped appearance of the articular surface (2/25 bones; see Figs. 1C and 4A). Atypically shaped phalanges were most often observed in DIV (11/25 bones) and DII (9/25 bones), with fewer seen in DI (3/25 bones) and DV (2/25 bones). No atypically shaped bones were observed in DIII. A GEE (negative binomial model) found age ($p = 0.002$), species ($p = 0.02$) and foot type ($p = 0.01$) to be statistically significant predictors of atypically-shaped phalanges, being more frequent in younger elephants, Asian elephants, and hind feet (20 bones in 12 hind feet vs five bones in five forefeet) in multivariate modelling. Sex was not significant ($p = 0.8$).

Phalangeal number varied between digits and feet. All African elephants had only the proximal phalanx in DI of their forefeet, and no phalangeal bones visible in DI of their

hind feet. The distal phalanx of DII was occasionally absent (2/10 African forefeet and 3/7 hind feet). The distal phalanx was always absent from DV in all African elephant feet. Subjectively, Asian elephants appeared to exhibit slightly more variability in phalangeal number. All Asian elephants lacked at least the middle phalanx in DI of their forefeet, however some also lacked the distal phalanx (9/18 Asian forefeet), and one foot lacked all phalanges in DI. In the hind feet of Asian elephants, some lacked only the distal phalanx from digit I (2/17 hind feet), some also lacked the middle phalanx (4/17), and most lacked all three (11/17). In DII, 1/17 hind feet was missing a middle phalanx and 1/17 was missing a distal phalanx. In DIII, 1/18 forefeet was missing a distal phalanx. In DIV, 4/18 forefeet were missing the distal phalanx and 1/18 forefeet was missing all three phalanges (suspected digital amputation, given the CT appearance). In DV, 3/18 forefeet and 11/17 hind feet were missing the middle phalanx (Fig. 4B), whilst 1/17 hind feet was missing both middle and distal phalanges.

## DISCUSSION

All elephants and almost all feet in this study were found with lesions likely to represent clinically important pathology. The elephants in our study are a biased population in this regard—though cause of death was not always clearly specified, it appears at least five of the 21 elephants died or were euthanised in part due to foot or joint problems. Despite this, our findings reinforce the longstanding concern that foot problems are frequent causes of morbidity and mortality in captive elephants (*Steel, 1885*; *Fowler, 2001*; *Luikart & Stover, 2005*; *Siegal-Willott, Alexander & Isaza, 2012*).

In addition to foot problems that are widely acknowledged in the literature on elephant pathologies (OA, infectious OA, osteitis, fractures and subluxation), we have observed remodelling of bones, enthesopathy, osseous cyst-like lesions, soft tissue mineralisation and hyperattenuating bone foci. We also found atypically shaped and absent phalanges, though any pathological significance of these features is unclear. Most of the elephant feet in this study had several pathological diagnoses (Table 3), supporting the notion that the different types of pathology have common causes, and/or that the establishment of one disease process may predispose elephants to developing others. For many types of pathology, multiple feet from the same elephant were affected, consistent with a generalised predisposition (e.g., husbandry, obesity; see also *Miller, Hogan & Meehan, 2016*) rather than singular cause. Most of our findings generally fall into three (sometimes overlapping) categories: lesions related to weight-bearing and loading of tissues, lesions related to ascending infection, and variable anatomy with unclear pathological significance.

Loading appears to have a significant influence on the development of pathology. A large proportion of the identified pathology was concentrated on the lateral three digits (remodelling, enthesopathy, osteitis, and infectious OA) or middle three digits (OA and osseous cyst like lesions); digits III and IV being the common denominator in both cases. The body weight of elephants is thought to be principally borne by the middle three digits (DII, DIII, and DIV) (*Siegal-Willott, Alexander & Isaza, 2012*), with the lateral three digits (DIII, DIV, DV) typically experiencing the greatest pressures during walking

(*Panagiotopoulou et al., 2012*). Contrary to expectations, we did not find the forelimbs to be significantly more affected by pathology than the hind limbs (*Hittmair & Vielgrader, 2000*), despite bearing a greater proportion of bodyweight (~60%; *Genin et al., 2010*). However, pressures on the forefeet are only higher in some instances and regions (*Panagiotopoulou et al., 2012*). Additionally, the digital cushions and predigits differ between fore and hind feet (*Weissengruber, 2006*; *Hutchinson et al., 2011*), and the limbs may be used differently in different styles of locomotion or other behaviours, potentially resulting in different patterns of loading between feet.

In OA, the link to increased or altered loading (via obesity or poor conformation) is fairly well established, though other factors (including trauma) may be involved (*Fowler, 2006*; *Siegal-Willott, Alexander & Isaza, 2012*). For other (putative) types of pathology, such as remodelling, enthesopathy and soft tissue mineralisation, the link to large or abnormal loads is hypothesised from other species. Enthesopathy in humans can be seen in degenerative, inflammatory or metabolic diseases (*Ruhoy, Schweitzer & Resnick, 1998*), and with aging (*Shaibani, Workman & Rothschild, 1993*). But animal models show that enthesopathy can also occur without tendon microtears or inflammation and may be an adaptive response to loading (*Benjamin, Rufai & Ralphs, 2000*). Remodelling and enthesopathy are both frequently observed in rhinos and thought to reflect tissue loading (*Regnault et al., 2013*; *Galateanu et al., 2013*; *Stilson, Hopkins & Davis, 2016*). The linear appearance and the location of soft tissue mineralisation in our elephants suggest that the digital flexor tendons are the affected structures. Mineralisation of the deep digital flexor tendon in horses has been observed as a response to chronic injury (*Dyson, 2003b*), and general mineralisation has been described as a feature of tendinopathy (tendon disease arising from overuse) and following trauma in other species (*O'Brien et al., 2012*). The magnitude of load experienced by structures may be a factor (especially in OA and remodelling, which both increase with increasing age and therefore presumably body weight), as might the type of loading; e.g., altered locomotion or long periods of standing. As elephants are both very large and long-lived, they may be more predisposed to loading-associated pathology and/or bone remodelling (perhaps including the variable sesamoid and phalangeal bone appearances described below) compared with other species. Indeed, as ossification of the foot and other limb bones tends to begin relatively late in elephants (*Hautier et al., 2012*) and their growth plates also tend to close late in life (uncertain and variable timing but roughly at 8–20 years of age; *Roth, 1984*; *Siegal-Willott et al., 2008*), the growth patterns of elephant feet (and perhaps limbs more generally) may leave them more vulnerable to accumulation of pathologies, although much more research is required to test this speculation.

Osteitis and infectious OA often result from spreading soft tissue infections, or penetration of a foreign object into the foot (*Fowler, 2006*). Our study found the proximal bones and joints to be more affected, compared to the distal and middle phalanges more often reported in other studies (*Fowler, 2006* citing *Gage, 1999* and *Hittmair & Vielgrader, 2000*); this apparent discrepancy might be best explained by variability and sample sizes in both cases.

We observed subluxation and fracture, which may result from trauma but may also sometimes be incidental findings (for example, fracture of the distal phalanx in elephants;

*Fowler, 2006*). Post-mortem fracture or manipulation of bones out of congruency also cannot be ruled out. Interestingly, we frequently observed multipartite distal phalanges that appear very similar to fractured phalanges but that we inferred to be a distinct entity, based on the lack of callus or bone reaction. The phalanges resembled the incompletely ossified distal phalanges observed radiographically in juvenile Asian elephants (*Siegal-Willott et al., 2008*). The affected elephants in our study were also all Asian (no African), and the distal phalanges of the lateral digits (DV and to a lesser degree, DIV) were most frequently observed to be multipartite. Like *Siegal-Willott et al. (2008)*, we found bipartite phalanges (called 'unilateral wing lucencies') more common that tripartite phalanges ('bilateral wing lucencies'). We observed multipartite distal phalanges in elephants up to 55 years old, and so it seems that the ossification centres of these bones may not always fuse with age (similar to multipartite sesamoids). We acknowledge that the distinction between fracture and a congenitally multipartite bone can be subtle (or even impossible with chronic fractures; *Morandi, 2012*), and that the pathological significance of either condition appears negligible in the distal phalanx.

It is important that veterinarians and radiologists are aware of such apparently normal anatomical variations and incidental lesions when evaluating pathology in the feet. Best-known amongst these is variable phalangeal number, especially in DI and DV (*Ramsay & Henry, 2001*; *Fowler, 2006*; *Hutchinson et al., 2008*; *Siegal-Willott, Alexander & Isaza, 2012*). Our data also support this longstanding observation of elephants, and confirm that digits II, III and IV generally have three phalanges (although exceptions existed, especially amongst Asian elephants). Atypically shaped phalanges are another source of anatomical variation observed in this study.

Sesamoid bones also had variable appearances—not only the proximal sesamoid bones (generally paired bones in other species but which may be fused or asymmetrical in elephants), but also the predigits. These false 'sixth toes' seem to be modified sesamoids that start out as cartilaginous rods but may later ossify (*Hutchinson et al., 2011*). In our elephant sample (with sample overlap from those of *Hutchinson et al., 2011*), the predigits ranged from completely non-ossified (visible as a hollow cartilaginous rod), to small and patchy regions of mineral attenuation, to large discrete pieces of bone, to long, elaborate and jointed structures curving around to the back of the foot. Within the same animal, the degree of mineralisation in pairs of forefeet or hind feet was consistent, but could vary between fore and hind limbs.

We found that Asian elephants showed a greater tendency towards ossification of the predigits. Presence of sesamoid bones at joints has been linked to increased OA by some studies (e.g., *Pritchett, 1984*; *Hagihara et al., 1993*), though not others (e.g., *Muehleman, Williams & Bareither, 2009*). The possible link to OA in humans has prompted the hypothesis that sesamoids may predispose joints to developing disease, or that both OA and sesamoids are linked by an underlying process (i.e., tendency for endochondral ossification; *Sarin et al., 1999*). Although we did not find significantly more OA in Asian compared to African elephants, we did find more enthesopathy, more sesamoid fusion, and multipartite distal phalanges (indicating multiple unfused ossification centres). Along with their greater predigit ossification, these findings lead us to speculate that Asian elephants might have

an increased tendency for endochondral ossification (in their distal limbs) than African elephants. This could explain some differences in disease prevalence and bone anatomy.

Of our findings, only the osseous cyst-like lesions and hyperattenuating regions do not clearly fit into the categories of lesions related to loading, infection, or incidental finding/variable anatomy. Osseous cyst-like lesions may be secondary to OA, osteochondrosis (particularly if subchondral), ischaemic necrosis, haemorrhage, or vascular malformation (*Carlson & Weisbrode, 2006*). Like our elephants, sex-based biases in cyst prevalence have been noted in humans (*O'Donnell, 2009*) and some other animals (*Craig, Dittmer & Thompson, 2016*). The hyperattenuating regions resemble enostoses (benign foci of dense bone), which are sometimes associated with lameness in horses (*Dyson, 2003a*). The cause is unknown, but contributing factors may include excess dietary calcium (*Carciofi & Do Prado Saad, 2008*).

## CONCLUSIONS

Though a small proportion of our elephants were previously known to have foot or joint problems, the generally high level of pathology found in our study highlights the need for continuing vigilance regarding elephant foot health. We should not be complacent with lack of lameness or externally apparent signs. A comprehensive evaluation of foot health in elephants should therefore include 'baseline' foot radiographs to establish the 'normal' anatomy for that individual, and annual assessment thereafter using radiographic protocols with standard views optimal for the detection of pathological lesions (*Mumby et al., 2013*). In addition, weight management, regular exercise, a clean and appropriate environment (with minimal time spent on hard surfaces; *Miller, Hogan & Meehan, 2016*), and other measures to prevent over-loading, injury and infection should not be overlooked.

## ACKNOWLEDGEMENTS

We thank Yu-mei Chang for statistical advice, and others who have helped on aspects of this project before including Louise Ash, Reshma Biljani, Stefania Danika, Zoe Hill, Sophie Jenkins, Charlotte Miller, Olga Panagiotopoulou, Sharon Warner, and members of the Structure & Motion Laboratory. We also are grateful to all of the suppliers of elephant specimens or images used here, including Thomas Hildebrandt, Guido Fritsch, ZSL Whipsnade Zoo, John Cracknell (Longleat Safari Park), and Drs Daniela Denk and Mark Stidworthy (IZVG Pathology, UK). Finally, we thank two anonymous reviewers for their helpful comments on a previous version of this manuscript.

### Funding

This research was funded by BBSRC grants BB/H002782/1 (to JRH and RW) and BB/C516844/1 (to JRH). The funders had no role in study design, data collection and analysis, decision to publish, or preparation of the manuscript.

## Grant Disclosures

The following grant information was disclosed by the authors:
BBSRC grants: BB/H002782/1, BB/C516844/1.

## Competing Interests

John R. Hutchinson is an Academic Editor for PeerJ.

## Author Contributions

- Sophie Regnault and Jonathon J.I. Dixon conceived and designed the experiments, performed the experiments, analyzed the data, wrote the paper, prepared figures and/or tables, reviewed drafts of the paper.
- Chris Warren-Smith conceived and designed the experiments, reviewed drafts of the paper, development of pathology categorisation.
- John R. Hutchinson conceived and designed the experiments, contributed reagents/materials/analysis tools, wrote the paper, reviewed drafts of the paper.
- Renate Weller conceived and designed the experiments, analyzed the data, contributed reagents/materials/analysis tools, wrote the paper, reviewed drafts of the paper.

## Animal Ethics

The following information was supplied relating to ethical approvals (i.e., approving body and any reference numbers):

The work involved cadaveric material and/or CT scans from vertebrate animals, euthanised for reasons unrelated to the study and subsequently donated to the RVC, and did not require ethical approval.

## Data Availability

The raw data has been supplied as a Data S1.

## Supplemental Information

Supplemental information for this article can be found online at http://dx.doi.org/10.7717/peerj.2877#supplemental-information.

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
