# Peer review of "Skeletal pathology and variable anatomy in elephant feet assessed using computed tomography"

_PeerJ, doi:10.7717/peerj.2877_

## Round 0.1 · original submission · Major Revisions

Both reviewers have made a number of valid comments, particularly with respect to the statistical analyses and the content of your Discussion. Please take these suggestions to heart in your revised manuscript and respond to each of them individually, stating exactly what changes you have made to the manuscript.

Reviewer 1 ·

Basic reporting

The present study investigated foot problems in elephants as they are a major cause of morbidity and mortality, but are thought to be underreported. The investigators examined 52 feet from 21 elephants using CT scans (a mix of African and Asian), describing both pathology and variant anatomy. In all elephants some pathology was observed in at least one foot. Common pathological changes included bone remodelling, enthesopathy, osseous cyst-like lesions, and osteoarthritis, with soft tissue mineralisation, osteitis, infectious osteoarthritis, subluxation, fracture and enostoses. Most feet had multiple categories of pathological change (81% with two or more diagnoses, versus 10% with a single diagnosis, and 9% without significant pathology). The pathological changes were concentrated over the middle/lateral digits, which bear most weight and have high peak pressures during loading. Remodeling and osteoarthritis were correlated with increasing age. Multipartite, missing and misshapen phalanges were common and apparently incidental findings. The proximal sesamoids are fused or absent, while the predigits were variably ossified. Our study reinforces the need for regular examination and radiography of elephant feet to monitor for pathology and as a tool for improving welfare.

Basic reporting:
The paper was straight forward with clear descriptions and locations of the pathologies throughout the paper. The introduction was thorough with discussion of the rationale of the study and the general problems encountered with examination of these elephant specimens in attempts to be aid in the overall health and well-being of elephants in captivity.

Experimental design

The methods section was in sufficient detail with descriptions and grading of the potential pathologies encountered in this study. Statistical analysis used was appropriate.

Validity of the findings

The Result section was presented in a clear descriptive manner as they described and tabulated each of the pathologies observed on CT scan of each foot. The data were presented in a clear and concise manner from those most frequently observed (bone remodeling) to those less frequently seen (fractures and high density bone areas). The data were presented in relationship to the frequency observed, elephant species and age and sex of the elephant.
The findings and conclusions are valid in terms of their presentations. Their conclusions are valid and clearly presented.
The discussion is presented in a reasonable manner and detail.

Additional comments

The discussion is presented in a reasonable manner and detail. However, this reviewer believes that it could be a more interesting discussion with perhaps greater discussion of the bone remodeling of the distal limb and foot structures as the elephant is unusual animal in that they live longer that most of the large ruminants and horses and they are much larger in size. The large size as they suggested would put more load on the digits (middle and lateral ones) and these were the most deformed. Is there any information in the cited references regarding the closing of the growth plates (I assume 5-8 years?). There is significant information in both humans and large domestic animals that their distal limb bones (ie. Loading structures) can continuously remodel throughout the person’s or animal’s life. There is comparative data in horses and horses that may be relevant to potentially compare or provide speculative hints in the elephant. The assumption would be that the greater weight during loading and the longevity might have greater effects upon the bone variations in the elephant than in other animals. In this discussion a short paragraph would be a nice addition to the manuscript. A similar small expansion in the discussion could be added in discussing certain pathologies of the distal limb: lines 290-296 approximately- they mention the possibility of weight as a factor in influencing the bone variables, but cite another paper stating that most weight in on the forelimb and that more variability and pathologies were seen in the hind limbs. (NOTE: line 293- hind limbs is two words). In other animals the front and hind limbs are used differently during movements and locomotion as well as the digital cushions are different between the front and hind feet. These two factors could explain why more changes were seen in the hind limbs? An added short speculation would be an interesting addition to the text in this regard.
While the authors did discuss the pathologies, this reviewer would appreciate a little more in depth discussion on a comparative basis which might be very helpful to the reader who might be interested in the elephant pathologies of the foot. Again a few sentences and references strategically placed would be helpful as the elephant is only one of many large vertebrates that have similar limb issues as the large domestic and wild mammals.
One added reference regarding entheseophytes is Michael Benjamin's work who is also in the UK. His laboratory has published many articles on entheseophytes. Some of their works in human and various animals would be very important to the elephant literature.

Reviewer 2 ·

Basic reporting

Please revise manuscript to adhere to PeerJ in-text citations and Reference guidelines. Currently does not conform.
-https://peerj.com/about/author-instructions/#reference-format
---For three or fewer authors, list all author names (e.g. Smith, Jones & Johnson, 2004). For four or more authors, abbreviate with ‘first author’ et al. (e.g. Smith et al., 2005).
---Example journal reference: Smith JL, Jones P, Wang X. 2004. Investigating ecological destruction in the Amazon. Journal of the Amazon Rainforest 112:368-374. DOI: 10.1234/amazon.15886.

Experimental design

No comment.

Validity of the findings

• A summary table detailing the physiology results needs to be included in this manuscript. The results section is difficult to follow and it may help the authors to identify where further statistical comparisons need to be made or support currently unsupported claims in the discussion.
• There are multiple occasions where qualifying statements such as “more frequent” or “often” are made that are not supported by statistical evidence. This study is limited by a small sample size, and there are twice as many Asian elephants than Africans. In and of itself, this is reasonable given the source of data and the number of cadaver elephants with radiographs available for study. What is not reasonable is to make statements directly comparing frequencies without accounting for the different sample sizes using basic statistical measures. This biases the conclusions a reader may make and influences statements included in the discussion.

Additional comments

Introduction
Line 89: “legitimate” zoos. Vague. Specify if you mean accredited institutions or other. Or remove term completely.

Methods
Line 106-113. For GEE, please describe if these models run as multi-variable neg bin regressions, or separate for each predictor. If multi-variable, was it forward/backward/or stepwise selection?

Later, in the results section (line 233), you mentioned GEE results within each species. Were these independent variables run as separate bi-variable models after you discovered that species was significant, or was species included with each independent variable as an interaction term? Include in methods section.

Were the models assessed for confounding variables and subsequently controlled for? For example, on multiple occasions you state that species was significant. However, you have a significantly older population of Asians than Africans, and skewed sex distribution that differs between species as well. Neither of these facts is currently discussed in MS. Without knowing if this was controlled for in the stats, the statistical conclusions could very well be biased and mask the effect of other potential predictors.

Results
Include a summary table with pathology results in the main publication. This section as written is hard to read and follow, a table would be helpful to guide through the different results. The supplemental/raw data is NOT sufficient in its current form (no clear titles, column headers are not intuitive, no guide for understanding cell highlights).

Note, I appreciate the breakdown of frequency comparisons between N elephants and N feet. Thank you for including those counts in this section.

Line 169-170: awkward phrasing. “For the affected elephants with multiple feet scanned (how many were these? 15?), OA was always observed in multiple feet (9/15 elephants) (This sounds like 9 out of 15 had OA in multiple feet and 6 had OA in a single foot).” I’m guessing that you mean to say that 15 animals had multiple feet scanned, and 9 of these had OA, and of those 9 that had it, OA was always present. Please clarify.

Line 171-175: 7% of all pathologies observed. How many? What are the frequency and total N pathologies? Include at each specific pathology section. Or detailed in a summary table.

Line 215: Awkward phrasing. Please either fix for consistency or remove the qualifier “often.” Here you compare by species as “always present in Africans…but often missing in Asians.” Then you state absent in 6/14 Asian elephants, which means it is present in 8/14 Asian elephants.

Line 222: What do you mean by absent? Not visible on scan? Or known to be not present on foot?

Line 227-228: Remove “as a trend.” Here, you are just reporting frequencies, not looking at the statistics, which is what I was expecting from the phrasing.

Line 257: Remove “slightly more variability.” Vague and misleading given lack of statistics here.

General in Methods: Several times (Line 226, 239, 251) statements are made comparing the two species that have different sample sizes and claiming that one is “more frequent” simply because the percentages differ. No stats are given to back up the “more frequent” claims, and given the difference in sample sizes (twice as many Asian elephants than African), I’m not sure you can make these frequency claims. Two sample t-test should be sufficient here. If they are different, then you have stats to back up your claim and make that comparison stronger, and if not, then you might want to rephrase your statements.

Discussion
Line 270: Could foot-related death have been assessed as a predictor?

Line 283: You suggest that the outcomes you observed are sometimes overlapping. Perhaps include results showing this? Such as how many unique pathologies a single elephant might bear.

Line 323: African data not shown in results section. Again, a table would help.

Line 364: Sex based biases. Here it sounds like you have a sex difference, but this is not shown in GEE results.

Conclusion
Was reason for death (or at least foot-specific vs not) controlled for or tested as a potential predictor? Based on your conclusions, it begs the question of whether the rate of pathology was higher in those euthanized for stated foot-health related reasons.

---

## Round 0.2 · Minor Revisions

I'm requesting a few additional changes:

Line 112: Please substitute "binomial" for "bin"
Line 142 and on: Please use either: 1) two significant digits for all P-values; e.g., 0.01 has one significant digit, while 0.24 has two significant digits; or 2) a consistent use of P-values to three decimal places (for example, Line 248 has P=0.70, Line 254 has P=0.308, etc.).

---

## Round 0.3 · accepted · Accept

And thank you for making the remaining minor edits.